# Zebrafish airinemes optimize their shape between ballistic and diffusive search

Sohyeon Park[1†], Hyunjoong Kim[2†], Yi Wang[3], Dae Seok Eom[1,3]*, Jun Allard[1,4,5]*

[1]Center for Complex Biological Systems, University of California, Irvine, Irvine, United States; [2]Center for Mathematical Biology, Department of Mathematics, University of Pennsylvania, Philadelphia, United States; [3]Department of Developmental & Cell Biology, University of California, Irvine, Irvine, United States; [4]Department of Physics and Astronomy, University of California, Irvine, Irvine, United States; [5]Department of Mathematics, University of California, Irvine, Irvine, United States

**Abstract** In addition to diffusive signals, cells in tissue also communicate via long, thin cellular protrusions, such as airinemes in zebrafish. Before establishing communication, cellular protrusions must find their target cell. Here, we demonstrate that the shapes of airinemes in zebrafish are consistent with a finite persistent random walk model. The probability of contacting the target cell is maximized for a balance between ballistic search (straight) and diffusive search (highly curved, random). We find that the curvature of airinemes in zebrafish, extracted from live-cell microscopy, is approximately the same value as the optimum in the simple persistent random walk model. We also explore the ability of the target cell to infer direction of the airineme's source, finding that there is a theoretical trade-off between search optimality and directional information. This provides a framework to characterize the shape, and performance objectives, of non-canonical cellular protrusions in general.

*For correspondence:
dseom@uci.edu (DSeokE);
jun.allard@uci.edu (JA)

†These authors contributed equally to this work

Competing interest: The authors declare that no competing interests exist.

## Editor's evaluation

This article studies statistical aspects of the role of long-range cellular protrusions called airinemes as means of intracellular communication. The authors use published data showing how airinemes approach a target cell and describe these movements with a mathematical model for an unobstructed persistent random walk. Beyond the specialized readers interested in modeling and airineme biology, this article will also be of interest to cell biologists and biophysicists interested in intracellular communication.

## Introduction

The question of optimal search — given a spatiotemporal process, what parameters allow a searcher to find its target with greatest success? — arises in many biological contexts for a variety of spatiotemporal processes. Examples of relevant processes include searchers moving by diffusion or random walks (*Lawley et al., 2020*; *Berg and Purcell, 1977*), Levy walks (*Fricke et al., 2016*), and ballistic motion (straight trajectories [*Bressloff, 2020*], which, e.g., arises in chromosome search by microtubules [*Holy et al., 1994*; *Paul et al., 2009*]), and combinations of these (*Berg, 1993*). Another type of motion is the persistent random walk (PRW), which has intermediate properties between diffusion and ballistic motion. PRWs have been studied in continuous space (*Schakenraad et al., 2020*; *Großmann et al., 2016*; *Khatami et al., 2016*) and on a lattice (*Tejedor et al., 2012*), have been used with variants to model cell migration (*Jones et al., 2015*; *Weavers et al., 2016*; *Harrison and Baker, 2018*), and are mathematically equivalent to worm-like chains, which have been used to study the search by

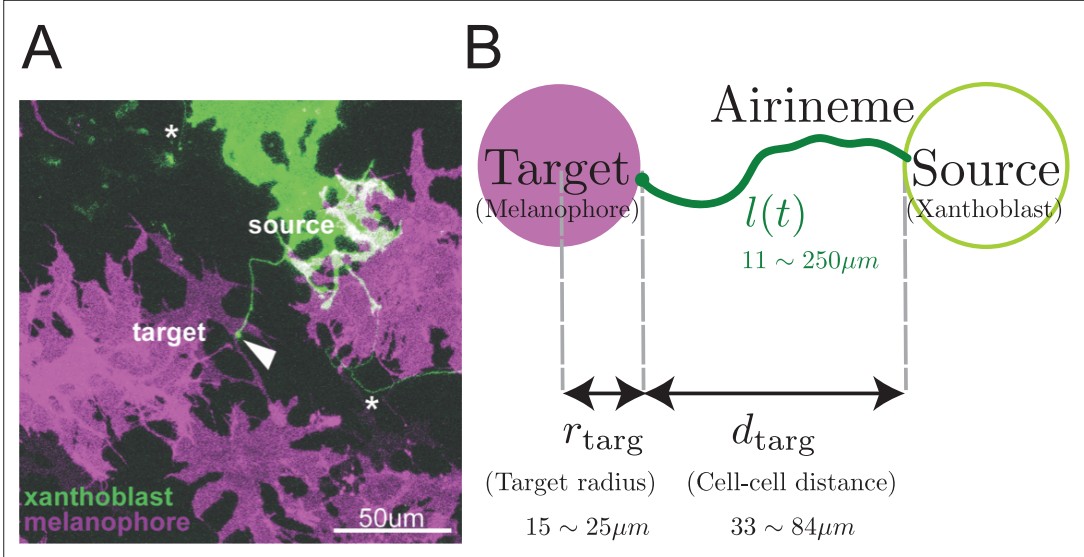

**Figure 1.** Airineme-mediated signaling between xanthoblast and target melanophore. (**A**) Multiple airinemes extend from xanthoblast (undifferentiated yellow pigment cell, green). Airineme makes successful contact (arrowhead) with melanophore cell (pigment cell, purple). Asterisks indicate airinemes from other sources. Scale bar: 50 μm. (**B**) Model schematic. A single airineme extends from the source (right, green circle) and searches for the target cell (left, purple circle). Target cell has radius $r_{\text{targ}}$ and has distance $d_{\text{targ}}$ away from the origin. The airineme's contour length at time $t$ is $l(t)$.

a polymer for a binding partner (***Mogre et al., 2020***). For all the above processes, optimality depends on parameters of the searcher (e.g., whether searchers operate individually or many in parallel; ***Schuss et al., 2019***; ***Lawley and Madrid, 2020***), the target(s), and the environment (***De Bruyne et al., 2020***; ***Bressloff, 2020***).

One example of a biological search process arises during organismal development, when cells must establish long-range communication. Some of this communication occurs by diffusing molecules (***Hu et al., 2010***; ***Govern and ten Wolde, 2012***; ***Bialek and Setayeshgar, 2008***; ***Endres and Wingreen, 2009***) like morphogens. However, recently, an alternative cell–cell communication mechanism has been revealed to be long, thin cellular protrusions extending tens to hundreds of micrometers (***Eom, 2020***; ***Yamashita et al., 2018***; ***Caviglia and Ober, 2018***; ***Sanders et al., 2013***; ***Bressloff and Kim, 2019***; ***Inaba et al., 2015***). These include cytonemes (***Kornberg and Roy, 2014***), tunneling nano-tubes (***Zurzolo, 2021***), tenocyte projections (***Subramanian et al., 2018***), and airinemes in zebrafish (***Volkening and Sandstede, 2018***; ***Volkening, 2020***; ***Eom and Parichy, 2017***; ***Eom et al., 2015***), shown in *Figure 1A*. One of the difficulties delaying their discovery and characterization is their thin, suboptical width, and the fact that they only form at specific stages of development (***Eom, 2020***; ***Yamashita et al., 2018***; ***Caviglia and Ober, 2018***).

Airinemes are produced by xanthoblasts (undifferentiated yellow pigment cells) and play a role in the spatial organization of pigment cells that produce the patterns on zebrafish skin (***Eom et al., 2015***; ***Eom and Parichy, 2017***; ***Eom, 2020***; ***Volkening and Sandstede, 2018***). Macrophages recognize a signal on xanthoblasts and begin dragging a protruding airineme from the xanthoblast as they migrate around the tissue, with the airineme trailing behind them. Airineme lengths have a maximum, regardless of whether they reach their target. If the tip complex reaches a target before this length, it recognizes target cells (melanophores) and the macrophage and airineme tip disconnect. The airineme tip contains the Delta-C ligand, which activates Notch signaling in the target cell. Due to experimental limitations on spatial and temporal resolution, the mechanism by which the airineme tip complex (which might include the entire macrophage) recognizes the target is still mysterious, as is the mechanism by which the macrophage hands off the airineme tip. It is also not known what other signals, if any, are carried by the airineme. If no target cell is found by the maximum length, the macrophage and airineme disconnect, and the airineme retracts. In the unrelated context of wound-healing, macrophages are recruited to the site of injury by detecting chemokines released by damaged cells

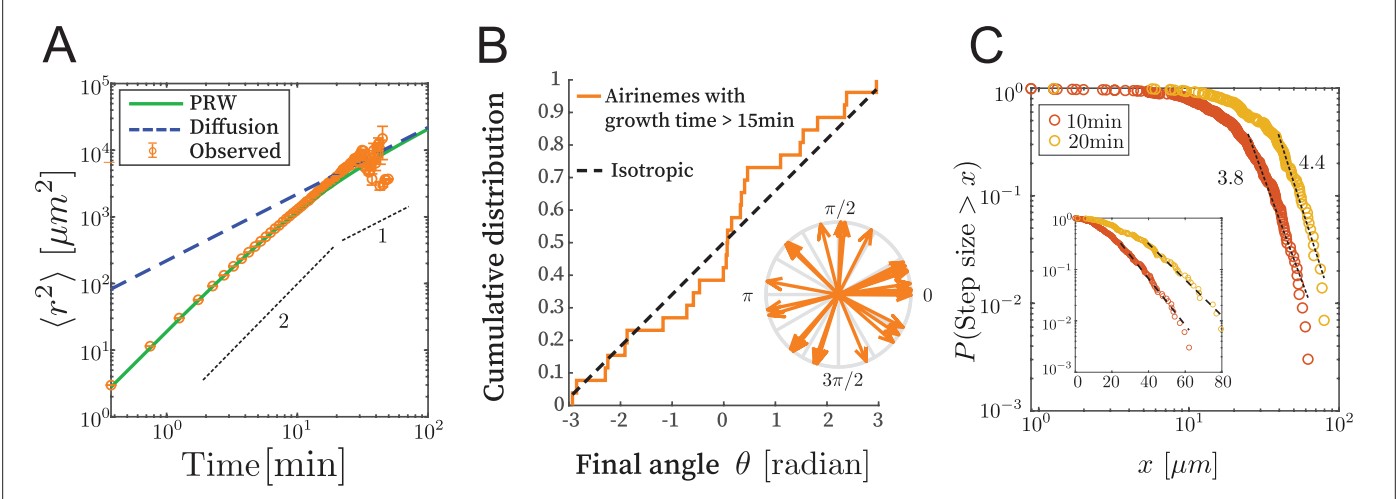

**Figure 2.** Airinemes are not consistent with simple random walk or Levy-type models. (**A**) Mean squared displacement (MSD) of airinemes. Airineme experimental MSDs have exponent $\approx 2$ (corresponding to slope in log–log plot) up to $\approx 15$ min, arguing against a simple random walk model. $N = 70$ airinemes. (**B**) For airinemes with growth periods longer than 15 min, the distribution of final angles, that is, the angle between the tangent vector at the airineme's origin and the tangent vector at its tip. This is consistent with isotropy, that is, a uniform distribution in $(-\pi, \pi)$, shown as a dashed line (Kolmogorov–Smirnov test, $N = 26$, p-value 0.37). The results of this test are insensitive to the cutoff length (see **Figure 2—source data 1**). (**C**) Step length complementary cumulative distribution function (CCDF). Levy models in 2D are characterized by CCDF tails with an exponent between 1 and 3. However, for two different time sampling intervals (10 and 20 min, from $N = 56$ airinemes), the tails of the distribution fit to an exponent greater than 3, and a continued downward curvature instead of a power-law, arguing against Levy models. Inset shows same data, same time sampling intervals, and same axes, but on semi-log plot.

The online version of this article includes the following source data and figure supplement(s) for figure 2:

**Source data 1.** Final angles are isotropic.

**Figure supplement 1.** Time evolution and final state are self-consistent.

or other immune cells. In contrast, macrophages pulling airinemes during development are not triggered by tissue damage or infections in zebrafish skin (**Eom et al., 2015**), and there is no experimental evidence that the airineme search process responds to any directional cues.

For diffusing cell–cell signals, dynamics are characterized by a diffusion coefficient. In contrast, cellular protrusions require more parameters to describe, for example, a velocity and angular diffusion, or equivalently a curvature persistence length. Here, we ask, what are these parameters, and what determines their values? We focus on airinemes where quantitative details have been measured (**Eom and Parichy, 2017**; **Eom et al., 2015**). We find that airineme shape is most consistent with a finite-length PRW model, and that the parameters of this model exhibit an optimum for minimizing the probability of finding a target (or, equivalently, the mean number of attempts). We compare this with another performance objective, the ability for the airineme to provide a directional cue to the target cell, and find that there is a theoretical trade-off between these two objectives. This work provides an example where a readily observable optimum appears to be obtained by a biological system.

## Results

### Airinemes are consistent with a finite PRW model

We examined time-lapse live-cell image data as described in **Eom and Parichy, 2017** and **Eom et al., 2015**. We confirmed that the time series and the final state are similar (**Figure 2—figure supplement 1**), meaning that the shape of the part of the airineme existing at time $t$ does not significantly change after time $t$, as the tip of the airineme continues to extend. This allows us to consider only the fully extended airineme and infer the dynamics, assuming airinemes extend with constant velocity v = 4.5 μm/min (**Eom et al., 2015**). This removes artifacts like microscope stage drift and drastically simplifies the analysis. We manually identified and discretized 70 airinemes into 5596 position vectors $r(t)$, and from these, computed the mean squared displacement (MSD) $\langle r^2 \rangle$. Random walks satisfy $\langle r^2 \rangle = 4Dt$.

However, the observed MSD, shown in *Figure 2A*, does not appear linear in $t$. We fit it to $\langle r^2 \rangle = \gamma t^\alpha$ and found the best-fit exponent $\alpha$ of 1.55 (90% CI in [1.50,1.61]), and indeed it appears that a single exponent is not appropriate across orders of magnitude. We therefore reject the simple random walk description.

Next, we consider Levy-type models such as those that have been used to describe animal optimal foraging (*Viswanathan et al., 2011*) and T cell migration (*Fricke et al., 2016*). These processes have a step size distribution whose tail has exponent between 1 and 3 in 2D (*Fricke et al., 2016*; *Viswanathan et al., 2011*), where step size is the displacement during a specified time interval. We revisit the time-series data (i.e., here we do not use the final state approximation) and compute a step length complementary cumulative distribution function (CCDF). For two time interval choices, shown in *Figure 2B*, the best-fit CCDF exponents are greater than 3 (for 10 min, exponent is 3.81 with 90% CI in [3.68,3.95]; for 20 min, exponent is 4.40 with 90% CI in [4.18,4.61]). Indeed, the CCDFs at two different time sampling intervals have continued downward curvature, indicating that a power-law description is inappropriate. We thus conclude that the process is not consistent with Levy-type models.

Finally, we consider a finite-length PRW. In this model, the tip of the airineme moves at constant speed $v$, while the direction undergoes random changes with parameter $D_\theta$, the angular diffusion coefficient. This parameter has units inverse minutes and roughly corresponds to the 'curviness' of the path. The dynamics are governed by *Equations 2–4*. A key observation from time-lapse imaging is that airinemes have a maximum length, after which the search process terminates if unsuccessful. Thus, our PRW model is finite-length, meaning that we assume the airinemes extend only up to $l_{\max} = 250\,\mu\text{m}$. This assumption yields a final length distribution (*Figure 3*, *Figure 3—figure supplement 1*) consistent with the observed distribution (*Eom et al., 2015*).

The airineme MSD fits the prediction of the PRW model, in *Equation 5*, up to time point around 15 min. Above this time, the PRW model is consistent with the data, although the low number of long airinemes in our data precludes a strong conclusion from MSD alone. We therefore took all airinemes whose growth time was greater than 15 min and plotted their final angle, that is, the angle between the tangent vector at their point of emergence from the source cell and the tangent vector at their tip. The PRW model predicts that, for long times $> 1/D_\theta$, the angular distribution should become isotropic. In *Figure 2B*, we find that the angular distribution is uniform, that is, isotropic (Kolmogorov–Smirnov test p-value 0.37, $N = 26$). Since there are relatively few data points, we repeated this analysis under various airineme selection criteria, which includes up to $N = 49$ airinemes, and in all cases found the final angular distribution to be consistent with uniformity (*Figure 2—source data 1*). (In *Figure 4*, we also check the autocorrelation function and further confirm consistency with the PRW model.) Taken together, the data favor the PRW model, which we use in the following analysis.

We also assume that airinemes operate independently as there is no evidence of airinemes communicating with each other during the search process. Furthermore, airinemes are generated at approximately 0.15 airinemes per cell per hour. Thus, the mean time between airineme initiations is $\approx 400\,\text{min}$, much larger than the time each airineme extends, which is 56 min. Note that many airinemes emanating from the same source cell may exist simultaneously, but most of the time only one airinemes is extending. Also, while the tissue surface is crowded, the airineme tips (which are transported by macrophages; *Eom et al., 2015*) appear unrestricted in their motion on the 2D surface, passing over or under other cells unimpeded (*Eom and Parichy, 2017*). We therefore do not consider obstacles in our model (although these have been studied in other PRW contexts; *Schakenraad et al., 2020*; *Khatami et al., 2016*; *Hassan et al., 2019*). This includes the source cell, that is, we allow the search process to overlie the source cell.

The target cell is modeled as a circle of radius $r_{\text{targ}} = 15 - 25\,\mu\text{m}$(*Eom et al., 2015*), separated from the source of the airineme by a distance $d_{\text{targ}} \approx 50\,\mu\text{m}$, as shown in *Figure 1B*. Including the position and size of the target, the model has five parameters, all of which have been measured (see *Table 1* and *Eom et al., 2015*; *Ryu et al., 2016*) except for $D_\theta$.

## Contact probability is maximized for a balance between ballistic and diffusive search

We performed simulations of the PRW model, testing different angular diffusion values for different values of cell-to-cell distance and target cell radius. For each parameter set, we measured the proportion of simulations that contacted the target. Specifically, contact is defined as the event in which the

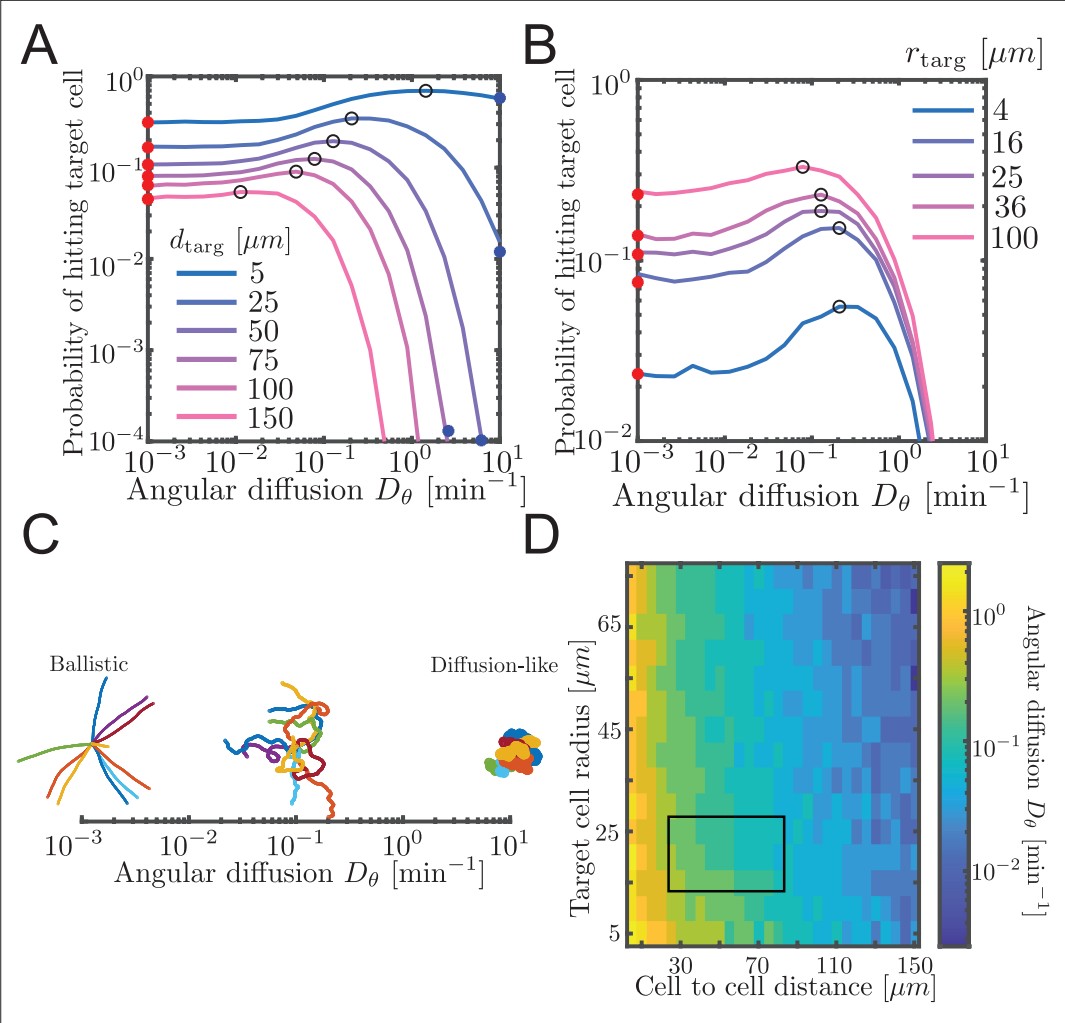

**Figure 3.** Probability to contact a target cell is maximized by a balance between ballistic search and diffusion-like search. (**A, B**) Simulated airineme target search for various $D_\theta$. The values of $D_\theta$ that leads to the highest search success probability are shown as black open circles. Theoretical values for ballistic limit from Equation 6 are shown as red circles. (**A**) Varying distances between the target and the source cell, while fixing the target cell radius at 25 μm. We validate simulation results with survival probability PDE at high $D_\theta$ from *Equation 7*, shown in blue circles. (**B**) Varying target cell radii while fixing the distance between the target and the source. For the biologically relevant parameter ($d_\text{targ} \sim 50\,\mu$m), optimal angular diffusion is around $D_\theta \sim 0.18\,\text{min}^{-1}$. (**C**) Qualitative behavior of airineme target search depends on $D_\theta$. (**D**) Optimal $D_\theta$ for a larger range of cell-to-cell distances and target cell radii. Rectangular region shows biologically relevant parameters.

The online version of this article includes the following figure supplement(s) for figure 3:

**Figure supplement 1.** Final lengths of airinemes.

**Figure supplement 2.** Simulated contact success shown as number of attempts instead of probability of contact.

tip of the growing airineme intersects with the target. In simulations, we assume contact has occurred when the airineme tip reaches within a distance $r_{targ}$ of the center of the target cell. We refer to $r_{targ}$ as the target cell radius. However, as discussed above, the mechanism by which contact is detected is unknown, and it could be that the airineme tip has a large effective spatial extent that includes some or all of the macrophage. Note again the search process is finite-length (otherwise in two dimensions would always eventually find the target). These contact probabilities are shown in *Figure 3A and B*. Equivalently, we plot the inverse, the mean number of attempts, in *Figure 3—figure supplement 2*.

We find that there exists an optimal angular diffusion coefficient that maximizes the chance to contact the target cell. The optimal value balances between ballistic and diffusion-like search. This has

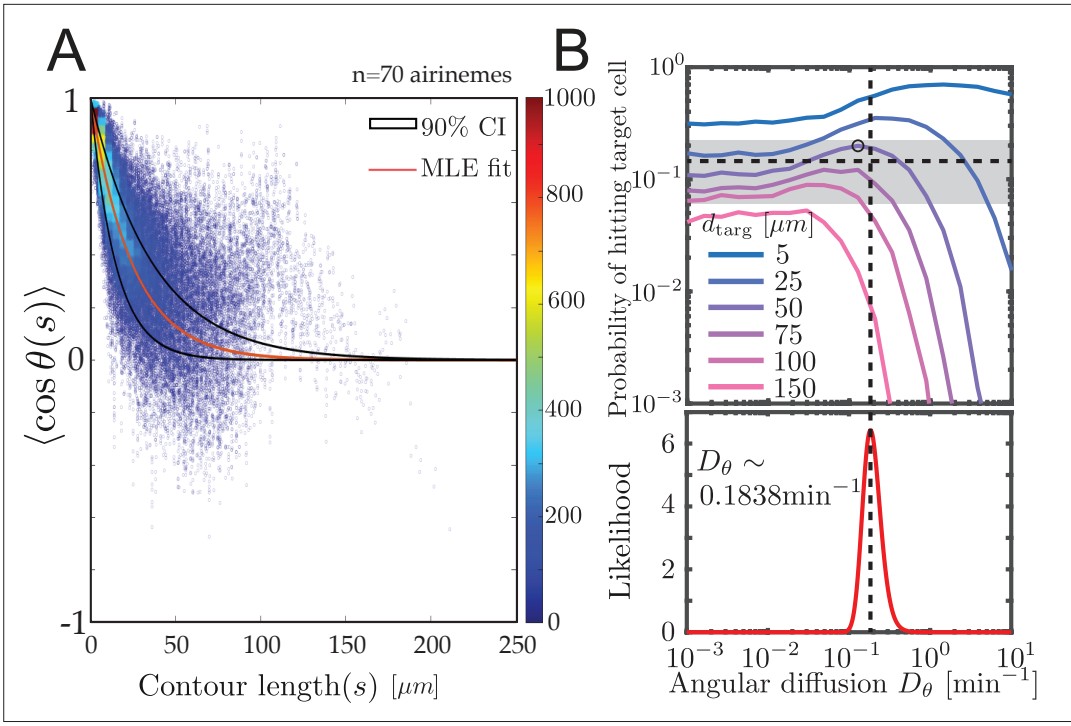

**Figure 4.** Experimental airineme curvature agrees with the optimal curvature. (**A**) Orientation autocorrelation function. We measure tangent angles $\cos(\theta)$ at 5596 points along 70 airinemes, and then compute the likelihood function (**B**, bottom) of $D_\theta$ fit to *Equation 1*. Best-fit curve is shown as red with a 90% confidence interval shown in black. Blue dots and heatmap were generated using a moving average with a window of 10 nearest data points and the heatscatter MATLAB function. (**B**) Bottom: we find that the best-fit airineme curvature from maximum likelihood estimation is $D_\theta = 0.1838\text{min}^{-1}$. We find that this value is similar to the $D_\theta$ that optimizes contact probability for the biologically relevant target cell distance $d_{\text{targ}} = 50\,\mu\text{m}$ (top). The experimentally observed probability of contact per airineme, center estimate (horizontal dashed line), and 90% confidence interval (gray area) are also shown.

The online version of this article includes the following figure supplement(s) for figure 4:

**Figure supplement 1.** Experimental data analysis method validation and data collection.

---

been previously shown for infinite, on-lattice PRWs (*Tejedor et al., 2012*) and worm-like chain models searching for binding partners (*Mogre et al., 2020*). We heuristically understand it as follows. When $D_\theta$ is small (*Figure 3C*, left), airinemes are straight and therefore move outward a large distance, which is favorable for finding distant targets. However, straight airinemes easily miss targets. On the other hand, for $D_\theta$ large (*Figure 3C*, right), the airineme executes a random walk. Random walks are locally thorough, so do not miss nearby targets, but the search rarely travels far. Thus, if the target cell is small or close, a diffusion-like search process is favored, but if the target cell is far or large, then a ballistic

**Table 1.** Parameter values.

| Symbol | Meaning | Estimate and value used | Source |
|---|---|---|---|
| $l_{max}$ | Maximum length of airineme | 250 µm | *Figure 3—figure supplement 1* |
| $v$ | Velocity of airineme growth | 4.5 µm (1.4–12 µm, N = 929) | *Eom et al., 2015* |
| $d_{targ}$ | Distance to target cell | 51 µm (33–84 µm, N = 70) | *Eom et al., 2015* |
| $r_{targ}$ | Target cell radius | 15–25 µm | *Ryu et al., 2016* |
| $D_\theta$ | Angular diffusion | $0.1838\,\text{rad}^2/\text{min}$ | Estimated here |

search is favored. We confirm this in *Figure 3D*, where we plot optimal $D_\theta$ over a large range of target cell radii and cell-to-cell distances. For the biologically relevant parameters (rectangular region in *Figure 3D*), a balance between ballistic and diffusion-like is optimal.

## Experimental airineme curvature is approximately optimal

In order to estimate the missing parameter $D_\theta$, we use the angular autocorrelation function

$$\langle \cos \theta(s) \rangle = \exp\left(-D_\theta s / v\right) \tag{1}$$

shown in *Figure 4*. We performed manual image analysis and maximum likelihood estimation to fit *Equation 1*, along with model convolution (*Figure 4—figure supplement 1*, *Gardner et al., 2010*, Materials and methods) to estimate uncertainty.

We find the maximum likelihood estimated persistence length of $12.24\,\mu\text{m}$, corresponding to an angular diffusion $D_\theta = 0.184\,\text{min}^{-1}$. Surprisingly, as shown in *Figure 4B*, this value matches our simulated optimal angular diffusion value for the biologically relevant parameter values. Moreover, in the experimental data, we find that the proportion making successful contact with target cells is $P_{\text{contact}} = 0.15$ (horizontal dashed line), from $N = 49$ airinemes with a 90% confidence interval in $[0.06, 0.24]$ (gray box in *Figure 4B*). This also agrees surprisingly well with the model prediction $p_{\text{contact}} \approx 0.185$.

## Directional information at the target cell

In some models of zebrafish pattern formation, the target cells receive directional information from source cells (*Eom, 2020*; *Volkening, 2020*), that is, the target cell must determine where the source cell is, relative to the target's current position. We explore the hypothesis that airineme contact itself could provide directional information since the location on the target cell at which the airineme contacts, $\theta_{\text{contact}}$ as shown in *Figure 5A*, is correlated with the direction of the source cell. Analogous directional sensing is possible by diffusive signals, where physical limits have been computed in a variety of situations (*Lawley et al., 2020*; *Berg and Purcell, 1977*).

We examined the contact angle distribution on the target cell. In *Figure 5B*, we show this distribution for three values of $D_\theta$: low (ballistic), high (diffusion-like), and the observed value we found above. The source cell is placed at $\theta_{\text{origin}} = 0$ without loss of generality (its initial direction is still chosen uniformly randomly). We show the distribution of contact angles on the target cell $p(\theta_{\text{contact}} | \theta_{\text{origin}} = 0)$ as both a radial histogram (top) and cumulative distribution (bottom). Interestingly, we find that the observed airineme parameters lead to a wide distribution of contact angles compared to both ballistic or diffusion-like airinemes.

To quantify the ability of the target cell to sense the direction of the source by arrival angle of a single airineme, we use Fisher information (*Fisher, 1997*), modified to take into account the probabilistic number of airinemes that a target cell receives, using *Equation 9*. Using this measure, we observe a minimum at intermediate $D_\theta$, shown in *Figure 5C*. We understand this intuitively as follows. For very straight airinemes, the allowed contact locations are restricted to a narrow range (a straight airineme can never hit the target's far side), resulting in high directional information. For high $D_\theta$, we initially expected low and decreasing directional information since there is more randomness. However, these are finite-length searches, and the spatial extent of the search process shrinks as $D_\theta$ increases. This leads to a situation where the tip barely reaches the target, and only at closest points (near $\theta_{\text{contact}} = 0$), resulting again in high directional information.

To compare with experimental observations, we attempted to measure the contact angle distribution of airinemes in contact with target cells. This is complicated by the highly noncircular shape of these cells, so we approximate the angle by connecting three points: the point on the source cell from which the airineme begins, the center of the nucleus of the target, and the point on the surface of the target where the airineme makes contact, as shown in *Figure 5—figure supplement 1*. We find a modified Fisher information of $5.7 \times 10^{-5}$, slightly smaller but similar in magnitude to the angle distribution predicted by the simulation.

## Trade-off between directional sensing and contact probability

By inspecting both $P_{\text{contact}}$ and directional information shown in *Figure 5D*, we find that there is a trade-off between the searcher's contact success and the target cell's directional sensing. Heuristically,

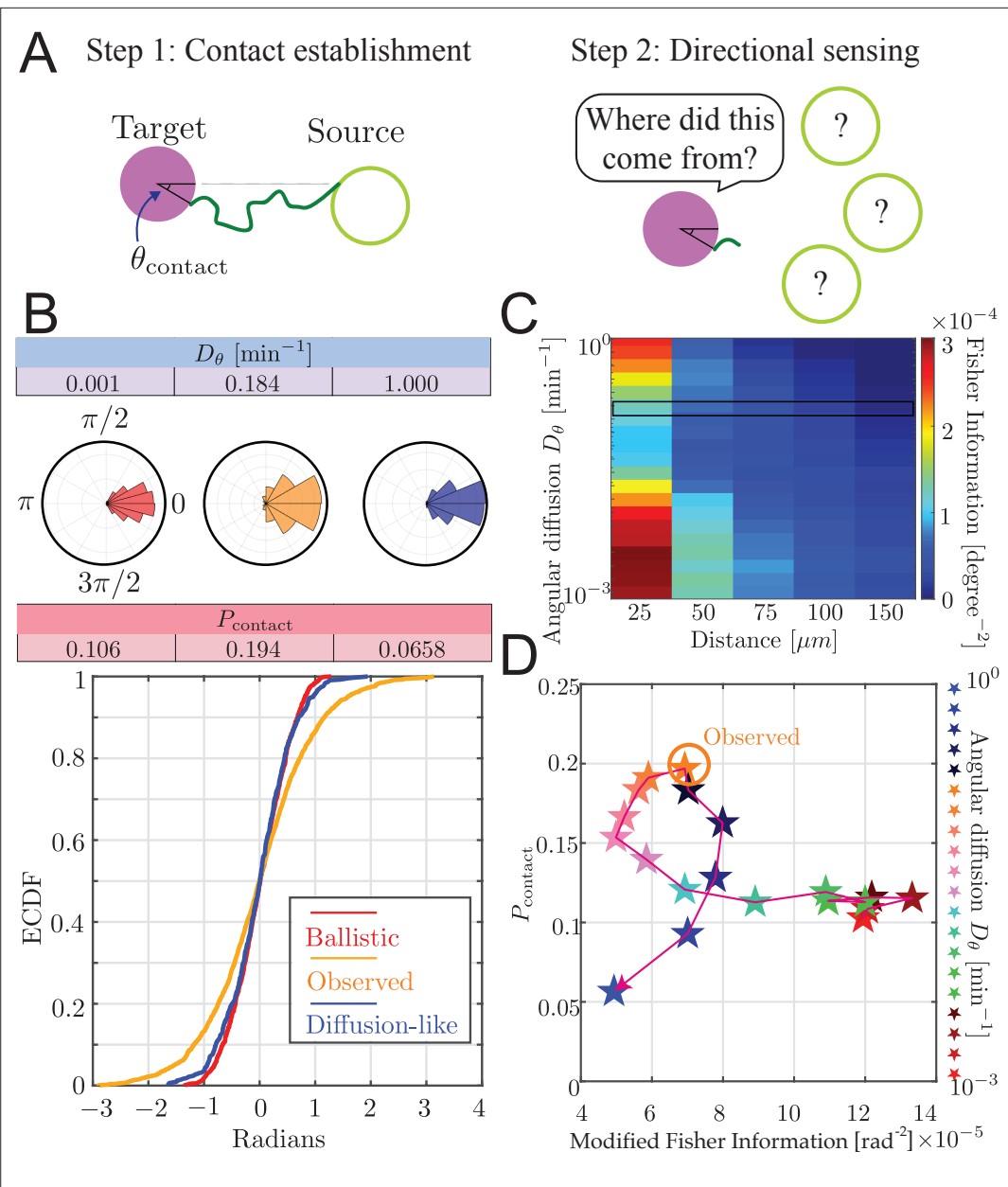

**Figure 5.** Trade-off between airineme directional sensing information and the probability of contacting the target cell. (**A**) Given that the source cell is located at $\theta = 0$, the distribution of angles at which the airineme contacts the target cell. (**B**) Given a source cell is located at $\theta = 0$, the distribution of angles at which the airineme contacts the target cell. Angle distributions with higher variance indicate poorer directional sensing. Three angular diffusion values (near ballistic limit, experimentally observed, and near diffusion limit) are shown. (**C**) Directional sensing accuracy, quantified using the modified Fisher information (FI) *Equation 9*, for different source-to-target distances and different ranges of angular diffusion values are tested, while fixing target cell radius at 25 μm. Black rectangular region shows the FI values for the observed airineme curvature. (**D**) Relationship between the contact probability $P_{contact}$ and FI for a range of $D_\theta$ (increasing with direction of arrow).

The online version of this article includes the following figure supplement(s) for figure 5:

**Figure supplement 1.** Experimental measurement of contact angle and directional information.

**Figure supplement 2.** Trade-off between directional sensing and contact probability for range of parameters.

this is because the two objectives prefer opposite variances. To maximize contact probability, variance should be maximal, taking full advantage of the surface of the target. On the other hand, to maximize directional information, the variance of contact angle should be minimized.

Interestingly, the experimentally observed $D_\theta$ is at a point where either increasing or decreasing its value would suffer one or the other objectives, a property known as Pareto optimality (**Alon, 2009**; **Barton and Sontag, 2013**). Note that this is also the $D_\theta$ value that maximizes search success, so the data is consistent with either conclusion that the curvature is optimized for search or it is optimized to balance search and directional information. In other words, in the case of zebrafish airinemes, there is no evidence that the shape of these protrusions sacrifices the goal of optimal search in order to achieve increased directional signaling. We wondered whether this is a general feature of search by PRW. The parametric curve in **Figure 5D** has a peculiar loop, the concave-down region giving rise to the Pareto optimum. In **Figure 5—figure supplement 2**, we show the parametric curve for a range of distances to the target $d_{targ}$ and target sizes $r_{targ}$. Note that we do not explicitly explore $l_{max}$, but since these plots have not been nondimensionalized, the parametric curve for a different $l_{max}$ can be obtained by rescaling the results shown. At low $d_{targ}$ (top row of **Figure 5—figure supplement 2**), the trade-off is amplified, and the parametric curve resembles bull's horns with two tips representing the smallest and largest $D_\theta$ in our explored range, pointing outward so the shape is concave-up. Intuitively, we understand this as follows: since the target is fairly close (relative to $l_{max}$), contact is easy. But the only way to get directional specification is by increasing $D_\theta$ to be very large, effectively shrinking the search range so it only reaches (with significant probability) the target at the near side at $\theta_{contact} = 0$. The parametric curve is concave-up, and there is no Pareto optimum. At high $d_{targ}$ (bottom row of **Figure 5—figure supplement 2**), the searcher either barely reaches, and does so at $\theta_{contact} = 0$, therefore providing high directional information, or $D_\theta$ is high, and the searcher fails to reach, and therefore also fails to provide directional information. So, there is no trade-off. At intermediate $d_{targ}$, the curve transitions from concave-up bull's horn to the no-trade-off diagonal line. Interestingly, it does so by bending forward, forming a loop, and closing the loop as the low-$D_\theta$ tip moves toward the origin. At these intermediate $d_{targ}$ values, the loop offers a concave-down region with a Pareto optimum.

## Discussion

As long-range cellular projections like airinemes continue to be discovered in multicellular systems, their mathematical characterization will become increasingly valuable, mirroring the mathematical characterization of diffusion-mediated cell–cell signals. We have measured the in situ shape of airinemes, and find agreeable fit to a finite, unobstructed PRW model, rather than Levy or diffusion-like motion. The mean square curvature, or equivalently the directional persistent length, is close to that which allows optimal search efficiency for target cells. Since airineme tip motion is driven by macrophages, our results have implications for macrophage cell motility, which is relevant in other macrophage-dependent processes like wound healing and infection (**Sun et al., 2019**; **Achouri et al., 2015**).

The growing catalog of non-canonical cellular protrusions (**Eom, 2020**; **Yamashita et al., 2018**; **Caviglia and Ober, 2018**; **Sanders et al., 2013**; **Bressloff and Kim, 2019**; **Inaba et al., 2015**; **Kornberg and Roy, 2014**; **Parker et al., 2017**; **Subramanian et al., 2018**; **Wang and Gerdes, 2015**) includes strikingly different shapes. For example, some tunneling nanotubes in cancer cells (**Parker et al., 2017**) are straight compared to airinemes. They also have different functions. For example, nanotubes in PC12 cells serve as conduits for organelles (**Wang and Gerdes, 2015**). This raises an intriguing possibility that different protrusions have a shape optimized for different functions. Besides search success probability and directional information, one obvious candidate for optimization is the efficient transport of signaling molecules after contact has been established (**Bressloff and Kim, 2018**; **Kim and Bressloff, 2018**). This might prefer shorter protrusion length, and therefore favor straight morphologies. In the future, it would be intriguing to compare all known non-canonical protrusions in light of the three performance objectives, and others.

Since the airineme tip's motion is linked to macrophage motion, these results also inform cell migration patterns. Variants of the PRW model have been found to describe cell migration (**Harrison and Baker, 2018**; **Weavers et al., 2016**). Specifically, a related model was found to accurately describe macrophages in zebrafish (**Jones et al., 2015**). Two observations from **Jones et al., 2015** are particularly relevant to our work: that macrophages in zebrafish demonstrate a mix of directional persistence

and randomness, and that their migration patterns adapt to circumstance (specifically, in their case, distance to wound and time since wounding).

Mathematically, the finite PRW process is equivalent to the worm-like chain model, for which exact formula have been derived for the tip location (*Spakowitz and Wang, 2005*; *Mehraeen et al., 2008*). The contact probability corresponds to a survival probability in the presence of an absorbing disk representing the target cell, and therefore an integral of the formulae in *Spakowitz and Wang, 2005*; *Mehraeen et al., 2008*. There is an opportunity to find analytic (asymptotic or exact) expressions for the contact probabilities, which would obviate the need for stochastic simulation.

The cell–cell interactions mediated by airinemes contribute to large-scale pattern formation in zebrafish, a subject of previous mathematical modeling (*Volkening and Sandstede, 2015*; *Volkening and Sandstede, 2018*; *Nakamasu et al., 2009*). Our results provide a contact probability per airineme, setting an upper bound on the ability of cells to communicate via this modality, which is itself a function of cell density (related to $d_{targ}$ in our notation). Thus, our results may inform future pattern formation models. In the reciprocal direction, these models may provide information about the distribution of target cells, which may significantly affect search efficiencies.

## Materials and methods

### Key resources table

| Reagent type (species) or resource | Designation | Source or reference | Identifiers | Additional information |
|---|---|---|---|---|
| Genetic reagent (*Danio rerio*) | Tg(tyrp1b: PALM-mCherry) | *Eom et al., 2015* | RRID:ZDB-TGCONSTRCT-141218-3 | |
| Recombinant DNA reagent | aox5: palmEGFP (plasmid) | *Eom et al., 2015* | RRID:ZDB-TGCONSTRCT-160414-1 | |
| Software | ImageJ | http://imagej.nih.gov/ij/ | RRID:SCR_003070 | |
| Software | MATLAB R2021b | http://www.mathworks.com/ | | |

### Zebrafish husbandry and maintenance

Adult zebrafish were maintained at 28.5°C on a 16 hr:8 hr light:dark cycle. Fish stocks of *Tg(tyrp-1b:palmmCherry)wp.rt11* (*McMenamin et al., 2014*) were used. Embryos were collected in E3 medium (5.0 mM NaCl, 0.17 mM KCl, 0.33 mM CaCl$_2$, 0.33 mM MgCl$_2$·6H$_2$O, adjusted to pH 7.2–7.4) in Petri dishes by in vitro fertilization as described in Westerfield with modifications (*Westerfield, 2004*). Unfertilized and dead embryos were removed 5 hr post-fertilization (hpf) and 1-day post-fertilization (dpf). Fertilized embryos were kept in E3 medium at 28.5°C until 5 dpf, at which time they were introduced to the main system until they were ready for downstream procedures. All animal work in this study was conducted with the approval of the University of California Irvine Institutional Animal Care and Use Committee (protocol #AUP-19-043) in accordance with institutional and federal guidelines for the ethical use of animals.

### Time-lapse and static imaging

The transgenic embryos, *Tg(tyrp1b:palmmCherry)*, were injected with the construct drive membrane-bound EGFP under the aox5 promoter to visualize airinemes in xanthophore lineages and melanophores (*Eom et al., 2015*). Zebrafish larvae of 7.5 SSL were staged following *Parichy et al., 2009* prior to explant preparation for ex vivo imaging of pigment cells in their native tissue environment as described by *Budi et al., 2011* and *Eom et al., 2012*. Time-lapse images, acquired at 5 min intervals for 12 hr, and static images were taken at ×40 (water-emulsion objective) on a Leica SP8 confocal microscope with resonant scanner.

### Model definitions, simulation, and analysis

In the finite-length PRW model, the position of the airineme tip at time $t$ is given by

$$\frac{dx}{dt} = v \cos(\theta(t))$$

(2)

$$\frac{dy}{dt} = v \sin(\theta(t)) \tag{3}$$

$$d\theta = \sqrt{2D_\theta} dW_t \tag{4}$$

where $W_t$ is a Wiener process, and $D_\theta$ is related to the directional persistence length $l_p$ in 2D by $D_\theta = v/2l_p$.

The MSD for PRWs is (**Wu et al., 2014**; **Sadjadi et al., 2020**)

$$\langle r^2 \rangle = 4l_p v t \left( 1 - \frac{2l_p}{vt} \left( 1 - \exp\left( -\frac{vt}{2l_p} \right) \right) \right). \tag{5}$$

To simulate this model, we use an Euler–Maruyama scheme with timestep $\Delta t \ll 1/D_\theta$, implemented in MATLAB (The MathWorks). To validate these simulations, at two limits of $D_\theta$, search contact probabilities can be solved analytically (**Figure 3A and B**, filled circles). First, the straight limit $D_\theta \to 0$. Suppose an airineme searches for the target cell centered at $(0,0)$ with radius $r_{\text{targ}}$, and the airineme emanates from a source at $(r_{\text{targ}} + d_{\text{targ}}, 0)$. Let $\phi$ be the angle between the hitting point on the target cell and the center line. Then,

$$\phi = \frac{\pi}{2} - \arccos\left( \frac{r_{\text{targ}}}{r_{\text{targ}} + d_{\text{targ}}} \right) \tag{6}$$

and $P_{\text{contact}} = \phi/\pi$. At the other limit, $l_P \ll d_{\text{targ}}$, the PRW is approximately equivalent to diffusion with coefficient $D = v^2/2D_\theta$. For a finite time $0 < t < l_{\text{max}}/v$ diffusive search process, the probability of hitting the target cell is $P_{\text{contact}} = 1 - S(r, t)$, where $S(r, t)$ denotes the survival probability, which evolves according to

$$\partial S/\partial t = D\nabla^2 S \tag{7}$$

with $S(r, t) = 0$ on the surface of the target cell. We solve this PDE and display results in **Figure 3A and B**, blue circles. For these validations, the $D_\theta$ values were chosen to fit the blue circles onto the plot.

## Image analysis and model fitting

In order to estimate uncertainty in our analysis method, we used model convolution (**Figure 4—figure supplement 1**, **Gardner et al., 2010**). Specifically, we first measured the experimental signal-to-noise ratio and point spread function. We then simulated airinemes with a ground-truth curvature value and convoluted the simulated images with a Gaussian kernel with the signal-to-noise ratio and point spread function measured from experimental data. Since there is a manual step in this analysis pipeline, independent analyses by five people were performed on both simulated data and experimental data. For simulated data, the difference between simulated and estimated $D_\theta$ was less than 7% in all cases and usually $\sim 2\%$.

The extracted data and analysis routines are available openly at: https://github.com/sohyeon-parkgithub/Airineme-optimal-target-search, (copy archived at swh:1:rev:366ad6e1e3e5061c0cf-395c8e3be784872903922; **Park, 2021**).

## Directional information

To measure the directional information that, stochastically, an airineme provides its target cell, we use the Fisher information,

$$I_1(\theta_{\text{origin}}) = \mathbb{E}\left( -\frac{\partial^2}{\partial \theta_{\text{origin}}^2} \log p\left( \theta_{\text{target}} | \theta_{\text{origin}} \right) \right). \tag{8}$$

This quantity can be intuitively understood by noting that, for Gaussian distributions, Fisher information is the inverse of the variance. So, high variance implies low information and low variance implies high information. If multiple independent and identically distributed airinemes provide information to the target cell, then the probability densities of each airineme multiply, and the Fisher information is $I(\theta_{\text{origin}}) = n_{hit} \cdot I_1(\theta_{\text{origin}})$, where $n_{hit}$ is the number of successful attempts, which is proportional to $P_{\text{contact}}$. Therefore, we define the modified Fisher information as

$$\text{Modifed FI} = I(\text{origin}) \cdot P_{\text{contact}}. \tag{9}$$

With this modification, a target cell that receives almost no airinemes will score low in directional information.

## Experimental measurement of directional information

Images capturing incidences of airinemes with membrane-bound vesicles extended from xanthoblasts, stabilizing on melanophores were captured and imported into ImageJ for angle analysis between (1) originating point of airinemes on xanthoblasts, (2) center of target cells (i.e., melanophores), and (3) docking site of airineme vesicles on target cells. Two intersecting lines were drawn as follows: (1) connect the originating point of an airineme on a xanthoblast with the center of the target melanophore to draw the first line, and (2) connect the center of the target melanophore to the docking site of the airineme vesicle on the target melanophore to draw the second line. The angle between the three points connected by the two intersecting lines was then generated automatically with the angle tool in ImageJ. Coordinates of each point and the corresponding angle were recorded with ImageJ and exported to an Excel worksheet for further analysis. Each angle was assigned a ± sign in the 180° system based on the relative location of the three points at the time of the airineme incident. The 0° line was defined as the line passing through the center of the target melanophore. Thus, a positive angle was assigned when the originating point of an airineme on a xanthoblast lies on the 0° line with the docking site of airineme vesicles on the target melanophore lies above the 0° line, and vice versa.

The extracted data and analysis routines are available openly at https://github.com/sohyeonparkgithub/Airineme-optimal-target-search.

## Acknowledgements

We thank Sean Lawley (University of Utah), Jay Newby (University of Alberta), and Yoichiro Mori (University of Pennsylvania) for valuable discussion. We acknowledge support from NSF CAREER award DMS-1454739 to JA, NIH R35GM142791 to DSE, NSF grant DMS 1763272 and two grants from the Simons Foundation (594598, QN and Math+X grant to the University of Pennsylvania).

## Additional information

### Funding

| Funder | Grant reference number | Author |
| --- | --- | --- |
| National Science Foundation | DMS-1454739 | Jun Allard |
| National Institutes of Health | R35GM142791 | Yi Wang<br>Dae Seok Eom |
| National Science Foundation | DMS 1763272 | Sohyeon Park<br>Jun Allard |
| Simons Foundation | 594598 | Sohyeon Park |
| Simons Foundation | Math+X U Penn | Hyunjoong Kim |

The funders had no role in study design, data collection and interpretation, or the decision to submit the work for publication.

### Author contributions

Sohyeon Park, Data curation, Formal analysis, Investigation, Software, Validation, Visualization, Writing – original draft, Writing – review and editing; Hyunjoong Kim, Conceptualization, Investigation, Mathematical model development, Software, Validation; Yi Wang, Formal analysis, Investigation; Dae Seok Eom, Conceptualization, Funding acquisition, Methodology, Project administration, Supervision, Writing – original draft, Writing – review and editing; Jun Allard, Conceptualization, Funding acquisition, Project administration, Supervision, Writing – original draft, Writing – review and editing

## Author ORCIDs

Hyunjoong Kim http://orcid.org/0000-0002-3534-2102
Yi Wang http://orcid.org/0000-0001-7409-4335
Dae Seok Eom http://orcid.org/0000-0002-0617-8788
Jun Allard http://orcid.org/0000-0002-2758-4515

## Ethics

All animal work in this study was conducted with the approval of the University of California Irvine Institutional Animal Care and Use Committee (Protocol #AUP-19-043) in accordance with institutional and federal guidelines for the ethical use of animals.

## Decision letter and Author response

Decision letter https://doi.org/10.7554/eLife.75690.sa1
Author response https://doi.org/10.7554/eLife.75690.sa2

---

# Additional files

## Supplementary files

• Transparent reporting form

## Data availability

Data and computational scripts are available in a repository mentioned in the manuscript (on GitHub) https://github.com/sohyeonparkgithub/Airineme-optimal-target-search, (copy archived at swh:1:rev:366ad6e1e3e5061c0cf395c8e3be784872903922).

The following dataset was generated:

| Author(s) | Year | Dataset title | Dataset URL | Database and Identifier |
|---|---|---|---|---|
| Park S, Kim H, Li Y, Eom DS, Allard J | 2021 | Airineme experimental data and codes | https://github.com/sohyeonparkgithub/Airineme-optimal-target-search | GitHub, GitHub |

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
