## [Editor Report]

This article studies statistical aspects of the role of long-range cellular protrusions called airinemes as means of intracellular communication. The authors use published data showing how airinemes approach a target cell and describe these movements with a mathematical model for an unobstructed persistent random walk. Beyond the specialized readers interested in modeling and airineme biology, this article will also be of interest to cell biologists and biophysicists interested in intracellular communication.

---

## [Decision Letter]

**Decision letter after peer review:**

Thank you for submitting your article "Zebrafish airineme shape is optimized between ballistic search and diffusive search" for consideration by *eLife*. Your article has been reviewed by 4 peer reviewers, one of whom is a member of our Board of Reviewing Editors, and the evaluation has been overseen by Naama Barkai as the Senior Editor. The following individual involved in review of your submission has agreed to reveal their identity: Elena F Koslover (Reviewer #1).

Essential revisions:

1) (Reviewer#3 and reviewer#2 – point 2) Some aspects of the biological situation under study must be explained better, as early as possible in the main text:

– The authors describe the characteristics of an airineme as it would be a signalling filopodia, e.g. a nanotube or a cytoneme, which sends out to target a cell.

An airineme is fundamentally different from a self-guided cellular protrusion since it is driven by a macrophage. Therefore, it is essential to focus on the "search-and-find" walk of the macrophage and not the passively dragged airineme. In the light of this discussion, it is not clear if statements like "allow the airineme to hit the target cell" are helpful as it would point towards an actively expanding protrusion like a filopodium. Furthermore, since the protrusion tip is directed by a macrophage, contact mean that the driving macrophage must contact the target cell and attached the airineme to it. So the airineme tip has a large spatial extent (the macrophage size), which will certainly affect the contact probability. The consequence of this for the probability of establishing contact must be discussed.

– In the current version of the paper, one must go to the material method section to understand that there is a maximal length for airinemes. For clarity it should be mention in the main text, because it is an important point of the discussion of Section 2.2 and Figure 3A. Indeed it is very well known that a 2D a diffusive walker will always find any target, which makes very surprising the Figure 3A until one understands that there is a maximal length in the model.

2) (Reviewer # 1, Reviewer #3 – point 1, Reviewer #4 ) One possibly surprising results is the fact that the diffusion coefficient is optimised both for finding the target, AND for finding the best compromised between finding the target and providing directional information, while the latter must necessarily require weaker diffusion. This necessitates more explanation. Is this true in general or does this rely on the particular range of parameter explored? Is it applicable to other systems involving a semiflexible structure reaching for a target or a moving agent executing a PRW?

3) Provide a point-by-point response to the reviewers' comments appended below.

*Reviewer #1 (Recommendations for the authors):*

I found this to be an interesting and well-written manuscript. Most of my recommendations are along the lines of suggestions for clarifications and further discussion placing this work into context.

1) Figure 2A is not very compelling in terms of the long-time scaling. Are there any other metrics that could be shown to bolster the case for approximately diffusive behavior at long times. Velocity correlation functions perhaps? Or step size distributions over long time intervals?

2) I was persistently confused when reading the paper (until I finally found it in the methods) about the definition of contact probability. It should be made clearer in the main text that this is the probability for a fixed length of airineme that somewhere along the length (or is it just at the tip?) it will intersect a circular target.

3) Some background context is provided in the intro and discussion linking the models here to previously explored stochastic processes that are described as persistent random walks. However, as I understand it, the persistent random walk is also mathematically equivalent to a wormlike chain in the polymer physics field. Given the authors are mostly exploring fixed structures of a mechanical object rather than particles moving through time, this is an analogy that could use further highlighting. There is extensive literature available on the distribution properties of wormlike chains. For example, I believe the distribution of angular source positions (used in calculating directional information) could be computed analytically using known wormlike chain distribution functions (such as in Spakowitz and Wang, 2005). In Mogre et al., Biophys J, 2020 very similar problems are discussed in the context of a wormlike chain polymer needing to contact a target with its tip, and the trade-off between a stiff, narrow path and a more meandering one depending on the polymer flexibility. The numerical calculations done in this paper are sufficient and reasonable for the problems addressed, but drawing connections to similar past work in the discussion may be helpful.

4) It would also be helpful to the reader to provide further context on the biological function and regulation of airinemes. In particular, the PRW model here necessarily assumes that the airineme tips grow in an unguided manner (as opposed to following potential signals that indicate target location). Is there any evidence that this is indeed the case? What is the functional role of the airinemes -- what is it they transport and how? Are there diffusing molecules that move through them? Motor-carried particles? Signaling waves? Do they exert mechanical forces on the target? I realize that incorporating transport processes along the airinemes is outside the scope of the calculations in this paper, but further discussion of these issues would be helpful to place the work in context.

5) It would also help to highlight which of the results encountered are generalizable to PRWs in many different systems, not just in airinemes. In particular, the fact that the optimal flexibility both maximizes contact probability and the trade-off between contact and directional information -- is this very specific to the particular length parameter or target size picked? Or is it a general feature of PRWs? If the former, what are the parameter criteria for which this relationship holds? Exploration of this would help future researchers looking to apply these results to biologically unrelated processes that show similar PRW behaviors.

*Reviewer #2 (Recommendations for the authors):*

The modelling suggests that the shape of the long-range projections can be established by macrophages and thus fits their random walk model. Indeed, such a mechanism would fit very nicely to previously published data describing the chemotaxis movement of macrophages in zebrafish wound healing (Phoebe et al., 2015; Inference of random walk models to describe leukocyte migration). The authors could explore this more in detail and propose a comparative analysis of macrophage movements in different contexts.

Airinemes seem to be protrusions transferring signals to a distant cell. This would be a similar aspect as for nanotubes and cytonemes, defined as signalling filopodia. There is now a good amount of literature on nanotubes from PC12 cells (e.g. structural components) and cytonemes in zebrafish (e.g. dynamics), which deliver the signal directly to a neighbouring cell. I believe the "search-and-find mode" could also be applied to these protrusions? The authors could use their model in the context of these actively extending signalling protrusions.

The authors mention that the "shape of an airineme does not change throughout extension". However, this is an unclear expression because the shape certainly refers also to the length.

*Reviewer #3 (Recommendations for the authors):*

My recommendation, following the list made in the public review are

1) Discuss the robustness of the conclusion regarding optimisation. hoe can the system be optimised both with respect to optimal contact and to the balance between optimal contact and optimal directionality information.

2) enhance the discussion regarding the biology of the system. What are you really modelling? The motion of a cellular protrusion whose velocity and persistence is related to its molecular constituent (cytoskeleton) or the motion of an entire cell (the macrophage quiding the protrusion).

3) Discuss the data in more detail, in particular how well they really agree with the model.

4) Discuss the assumption of the model more precisely, in particular regarding directional information.

*Reviewer #4 (Recommendations for the authors):*

I have only a few remarks that could be taken into account to improve clarity of the manuscript.

In the current version of the paper, one must go to the material method section to understand that there is a maximal length for airinemes. For clarity it should probably be better to mention it in the main text, because it is an important point of the discussion of Section 2.2 and Fig 3A. Indeed it is very well known that a 2D a diffusive walker will always find any target, which makes very surprising the Figure 3A until one understands that there is a maximal length in the model.

- Again for clarity it could be useful to present Fig1B also in semi-log scales since this type of curved lines in log-log scales may simply be exponentials. Identifying an exponential law for the step size distribution would certainly lead to a rejection of Levy type walks.

Please also define clearly what is the "best fit exponent" (Section 2.1, first paragraph) : which exponent is it (I guess it is the exponent in the MSD) ? Also what is the step size shown on Fig 2B (is it the distance travelled during a specific time ?)

- To avoid any confusion, it would be useful to draw Fig5A with an airineme that is not perpendicular to the cell surface, so that there is no confusion between the angle that the airineme's tip makes with the cell surface, and the contact angle.

---

## [Author Response]

Essential revisions:1) (Reviewer#3 and reviewer#2 – point 2) Some aspects of the biological situation under study must be explained better, as early as possible in the main text:– The authors describe the characteristics of an airineme as it would be a signalling filopodia, e.g. a nanotube or a cytoneme, which sends out to target a cell.An airineme is fundamentally different from a self-guided cellular protrusion since it is driven by a macrophage. Therefore, it is essential to focus on the "search-and-find" walk of the macrophage and not the passively dragged airineme. In the light of this discussion, it is not clear if statements like "allow the airineme to hit the target cell" are helpful as it would point towards an actively expanding protrusion like a filopodium. Furthermore, since the protrusion tip is directed by a macrophage, contact mean that the driving macrophage must contact the target cell and attached the airineme to it. So the airineme tip has a large spatial extent (the macrophage size), which will certainly affect the contact probability. The consequence of this for the probability of establishing contact must be discussed.

We have added a new paragraph in the Introduction emphasizing the role of the macrophage, and we have changed the language throughout the text. In particular, we want to remove “agency” from the airineme, since it is indeed moving with the macrophage. In the mathematical sections, we opt for the neutral phrase “search process”.

We have also clarified that, in the biological system, the details of contact are unclear (e.g., what mechanism in the macrophage-airineme-vesicle is responsible for distinguishing the target cell). Therefore, in the model, we have clarified that contact is declared when the airineme tip arrives at a distance r_targ from the center of the target cell, and this critical distance might be larger than the size of the target cell, since it might include part or all of the macrophage.

– In the current version of the paper, one must go to the material method section to understand that there is a maximal length for airinemes. For clarity it should be mention in the main text, because it is an important point of the discussion of Section 2.2 and Figure 3A. Indeed it is very well known that a 2D a diffusive walker will always find any target, which makes very surprising the Figure 3A until one understands that there is a maximal length in the model.

The finite length (after which the search process terminates if unsuccessful) is now discussed in the Introduction, and again in the first Results section, referring to supplemental Figure S4. The Reviewers’ statement, that two-dimensional diffusion is recurrent so always successful, is very important. So, in this version, we state this fact about 2d diffusion explicitly, and repeat the finite-length property upon first mention of search success probability.

2) (Reviewer # 1, Reviewer #3 – point 1, Reviewer #4 ) One possibly surprising results is the fact that the diffusion coefficient is optimised both for finding the target, AND for finding the best compromised between finding the target and providing directional information, while the latter must necessarily require weaker diffusion. This necessitates more explanation. Is this true in general or does this rely on the particular range of parameter explored? Is it applicable to other systems involving a semiflexible structure reaching for a target or a moving agent executing a PRW?

The Reviewer’s question is an excellent question: Is the trade-off between contact and directional information a general property of searchers that obey persistent random walks? To address this question, we now include the analysis previously contained in Figure 5D, but for a full parameter space exploration. This is done in new Figure 5 Supplemental Figure 1. In doing so, we found fascinating behavior that sheds some light on the loop in Figure 5D.

At low d_targ, the trade-off is amplified, and the parametric curve resembles bull's horns with two tips representing the smallest and largest D_theta in our explored range, pointing outward so the shape is concave-up. Intuitively, we understand this as follows: since the target is fairly close (relative to l_max), contact is easy. The only way to get directional specification is by increasing D_theta to be very large, effectively shrinking the search range so it only reaches (with significant probability) the target at the near side (“3-o-clock'' in Figure 5A). At low d_targ, the parametric curve is concave-up, and there is no Pareto optimum.

At high d_targ, the searcher either barely reaches (when D_theta is high), and does so at 3-o-clock, therefore providing high directional information, or D_theta is low, and the searcher fails to reach, and therefore also fails to provide directional information. So, at high d_targ, there is no trade-off.

At intermediate d_targ, the curve transitions from concave-up bull's horn to the no-tradeoff line. To our surprise, it does so by bending forward, forming a loop, and closing the loop as the low-D_theta tip moves towards the origin. At these intermediate d_targ values, the loop offers a concave-down region with a Pareto optimum.

So, to answer the specific question of the Reviewers: No, the Pareto optimum is not a general feature of persistent random walk searchers. It only exists in a particular parameter regime, sandwiched between a regime where there is a strict trade-off with no Pareto optimum and a regime in which there is no trade-off.

All of these results are now discussed in the main text.

(Note that although we do not explicitly explore lmax, since these plots have not been nondimensionalized, the parametric curve for a different lmax can be obtained by rescaling the results).

Reviewer #1 (Recommendations for the authors):I found this to be an interesting and well-written manuscript. Most of my recommendations are along the lines of suggestions for clarifications and further discussion placing this work into context.1) Figure 2A is not very compelling in terms of the long-time scaling. Are there any other metrics that could be shown to bolster the case for approximately diffusive behavior at long times. Velocity correlation functions perhaps? Or step size distributions over long time intervals?

See above discussion of long-time behavior, repeated here for convenience:

To reiterate the comment: the MSD analysis allows us to reject the simple random walk model, and it is consistent but alone is not strongly supportive of the PRW model, especially at high tau of around 15 minutes (long lengths of around 65 microns). As the Reviewer points out, this is due to low numbers of long airinemes.

This prompted us to investigate the long-length data using multiple analysis approaches. In the new manuscript, new Figure 2B, we took all airinemes whose growth time was greater than 15 min, and plotted their final angle, i.e., the angle between the tangent vector at their point of emergence from the source cell and the tangent vector at their tip. At long times, the PRW model predicts that, for long times >1/D_theta, the angular distribution should become isotropic.

In new 2B, we find that the angular distribution is uniform, i.e., isotropic, using a Kolmogorov-Smirnov test (p-value 0.37, N=26).

Since there are relatively few data points, we repeated this analysis under various airineme selection criteria, and in all cases found the final angular distribution to be consistent with uniformity (new Supplemental Data Figure 1). For example, if we set the threshold at 10min, which includes up to N=49 airinemes, the Kolmogorov-Smirnov test against a uniform angular distribution gives a p-value of 0.32.

We here add a few additional notes

– Note that there is significantly less data used in this test than in the MSD analysis or the autocorrelation function maximum likelihood analysis. In order to perform a hypothesis test, we wanted to be sure that the data points are independent, so we take only one from each airineme (unlike MSD and autocorrelation analyses, for which we take every interval of a particular length, whether in the same airineme or not.)

– Finally, although the >10min KS test has more data than the >15min KS test (N=49 compared to N=26), we have chosen to present the >15min KS test in the Main Text. As we mentioned above, the conclusion is unchanged for >10min (see Supporting Data). The reason is that >15min is the first test we ran to check angular distribution against a uniform (-pi,pi) distribution, and we did not want to bias our testing.

Taken together, the data are even more strongly supportive of the PRW model. We are grateful for the Reviewer in encouraging us to further explore the high-time data.

2) I was persistently confused when reading the paper (until I finally found it in the methods) about the definition of contact probability. It should be made clearer in the main text that this is the probability for a fixed length of airineme that somewhere along the length (or is it just at the tip?) it will intersect a circular target.

We have clarified the definition of contact when it is first discussed in Results. Since any already-created section of airineme does not move significantly after its creation (see Figure 1 Supplemental Figure 1), contact can only occur at the tip. (We assume in the model that, if contact had occurred somewhere behind the tip, then the airineme would have stopped growing).

3) Some background context is provided in the intro and discussion linking the models here to previously explored stochastic processes that are described as persistent random walks. However, as I understand it, the persistent random walk is also mathematically equivalent to a wormlike chain in the polymer physics field. Given the authors are mostly exploring fixed structures of a mechanical object rather than particles moving through time, this is an analogy that could use further highlighting. There is extensive literature available on the distribution properties of wormlike chains. For example, I believe the distribution of angular source positions (used in calculating directional information) could be computed analytically using known wormlike chain distribution functions (such as in Spakowitz and Wang, 2005). In Mogre et al., Biophys J, 2020 very similar problems are discussed in the context of a wormlike chain polymer needing to contact a target with its tip, and the trade-off between a stiff, narrow path and a more meandering one depending on the polymer flexibility. The numerical calculations done in this paper are sufficient and reasonable for the problems addressed, but drawing connections to similar past work in the discussion may be helpful.

Thank you for this suggestion. We became aware of the optimum in persistence length reported in Mogre et al., 2020 after submission, and we are happy to now include it in the Intro, and when we report the optimum in Results. We also have added a paragraph in Discussion referring to the analytic solutions found in Spakowitz 2005 and Mehraeen et al., 2007.

4) It would also be helpful to the reader to provide further context on the biological function and regulation of airinemes. In particular, the PRW model here necessarily assumes that the airineme tips grow in an unguided manner (as opposed to following potential signals that indicate target location). Is there any evidence that this is indeed the case? What is the functional role of the airinemes -- what is it they transport and how? Are there diffusing molecules that move through them? Motor-carried particles? Signaling waves? Do they exert mechanical forces on the target? I realize that incorporating transport processes along the airinemes is outside the scope of the calculations in this paper, but further discussion of these issues would be helpful to place the work in context.

The following is some background on airinemes, including what is known and what remains unknown. We have incorporated a shortened version of this as a new paragraph in the introduction.

Airinemes are produced by xanthophore cells (also called yellow pigment cells) and play a role in the spatial organization of pigment cells that produce the patterns on zebrafish skin. Xanthophores have bleb-like structures at their membrane, and those blebs are the origin of the airineme vesicles at the tip. Those blebs express phosphatidylserine (PtdSer), an evolutionarily conserved ‘eat-me’ signal for macrophages. Macrophages recognize the blebs, ‘nibble,’ and ‘drag’ as they migrate around the tissue and the filaments trailing and extending behind. Airineme lengths have a maximum, regardless of whether they reach their target. If the airineme reaches a target before this length, the airineme tip complex recognizes target cells (melanophores) and the macrophage and airineme tip disconnect.

Regarding the specific question about the mechanism by which airinemes functionally enact cell-cell communication: The airineme tip contains the Δ-C ligand, which activates Notch signaling in the target cell. The mechanism by which a macrophage hands off the airineme tip is still mysterious, due to temporal and spatial resolution limits. It is also known what other signals, if any, are carried by the airineme. If no target cell is found by the maximum length, the macrophage and airineme disconnect, and the airineme the extension switches to retraction. Thus, macrophages do not keep dragging the airineme vesicles until they find the target melanophores. However, how macrophages determine when to engulf the untargeted airineme vesicles is not understood.

Regarding the specific question about chemotactic signals: In a different context, during wound-healing, macrophages are recruited to the site of injury by detecting chemokines released by damaged cells or other immune cells. However, airineme pulling macrophage behaviors are not triggered by tissue damage or infections in zebrafish skin, and there is no experimental evidence that they respond to any directional cues.

Indeed, this last statement (that there is no experimental evidence of directional cue in the airineme search process) is further bolstered by the new Figure 2B, which shows a uniform long-time angular distribution of tip orientation.

5) It would also help to highlight which of the results encountered are generalizable to PRWs in many different systems, not just in airinemes. In particular, the fact that the optimal flexibility both maximizes contact probability and the trade-off between contact and directional information -- is this very specific to the particular length parameter or target size picked? Or is it a general feature of PRWs? If the former, what are the parameter criteria for which this relationship holds? Exploration of this would help future researchers looking to apply these results to biologically unrelated processes that show similar PRW behaviors.

See above discussion of generality of theoretical tradeoff, repeated here for convenience:

The Reviewer’s question is an excellent question: Is the trade-off between contact and directional information a general property of searchers that obey persistent random walks? To address this question, we now include the analysis previously contained in Figure 5D, but for a full parameter space exploration. This is done in new Figure 5 Supplemental Figure 1. In doing so, we found fascinating behavior that sheds some light on the loop in Figure 5D.

At low d_targ, the trade-off is amplified, and the parametric curve resembles bull's horns with two tips representing the smallest and largest D_theta in our explored range, pointing outward so the shape is concave-up. Intuitively, we understand this as follows: since the target is fairly close (relative to l_max), contact is easy. The only way to get directional specification is by increasing D_theta to be very large, effectively shrinking the search range so it only reaches (with significant probability) the target at the near side (“3-o-clock'' in Figure 5A). At low d_targ, the parametric curve is concave-up, and there is no Pareto optimum.

At high d_targ, the searcher either barely reaches (when D_theta is high), and does so at 3-o-clock, therefore providing high directional information, or D_theta is low, and the searcher fails to reach, and therefore also fails to provide directional information. So, at high d_targ, there is no trade-off.

At intermediate d_targ, the curve transitions from concave-up bull's horn to the no-tradeoff line. To our surprise, it does so by bending forward, forming a loop, and closing the loop as the low-D_theta tip moves towards the origin. At these intermediate d_targ values, the loop offers a concave-down region with a Pareto optimum.

So, to answer the specific question of the Reviewers: No, the Pareto optimum is not a general feature of persistent random walk searchers. It only exists in a particular parameter regime, sandwiched between a regime where there is a strict trade-off with no Pareto optimum and a regime in which there is no trade-off.

All of these results are now discussed in the main text.

(Note that although we do not explicitly explore lmax, since these plots have not been nondimensionalized, the parametric curve for a different lmax can be obtained by rescaling the results).

Reviewer #2 (Recommendations for the authors):The modelling suggests that the shape of the long-range projections can be established by macrophages and thus fits their random walk model. Indeed, such a mechanism would fit very nicely to previously published data describing the chemotaxis movement of macrophages in zebrafish wound healing (Phoebe et al., 2015; Inference of random walk models to describe leukocyte migration). The authors could explore this more in detail and propose a comparative analysis of macrophage movements in different contexts.

We agree completely that this is an important connection to make. We have added a paragraph in Discussion putting this work in the context of several previously published work describing cell migration, some of which use variants of persistent random walks. The data and analysis by Phoebe Jones et al., is particularly relevant because it studies macrophages in zebrafish. Unfortunately, while both models are qualitatively similar, having persistence and randomness, a direct comparison is not possible because of differences in the models (specifically, they report that their parameter “w” is strongly correlated with their parameter “b”, which is the parameter that would be comparable with our l_P or D_theta). Nonetheless, two observations from Jones et al., are intriguing in relation to our work: that macrophages in zebrafish demonstrate persistence, and that their migration patterns adapt to circumstance (specifically in their case, distance to wound and time since wounding).

Airinemes seem to be protrusions transferring signals to a distant cell. This would be a similar aspect as for nanotubes and cytonemes, defined as signalling filopodia. There is now a good amount of literature on nanotubes from PC12 cells (e.g. structural components) and cytonemes in zebrafish (e.g. dynamics), which deliver the signal directly to a neighbouring cell. I believe the "search-and-find mode" could also be applied to these protrusions? The authors could use their model in the context of these actively extending signalling protrusions.

This is an intriguing project idea. And, indeed, we are indeed planning to do a “comparative biology” study of the morphology (shape, length) of all noncanonical protrusions for which we can collect or find data. This raises the appealing possibility that different protrusions have a shape optimized for different functions, besides search success probability and directional information. One obvious candidate functional objective – that might apply especially to PC12 nanotubes – is the efficient transport of signaling molecules after contact has been established (see, e.g., Bressloff and Kim 2018, and Kim and Bressloff 2018). The transport time would decrease monotonically with protrusion length (in other words, it would favor straight protrusions). We are in the process of seeking resources for such a study. We believe it is outside the scope of the current manuscript. However, we have an extended, edited paragraph in Discussion with the exposition of this idea.

The authors mention that the "shape of an airineme does not change throughout extension". However, this is an unclear expression because the shape certainly refers also to the length.

Good point. We now use language, “the shape of the part of the airineme existing at time t does not significantly change after time t, as the tip of the airineme continues to extend.”

Reviewer #3 (Recommendations for the authors):My recommendation, following the list made in the public review are1) Discuss the robustness of the conclusion regarding optimisation. hoe can the system be optimised both with respect to optimcal contact and to the balance between optimal contact and optimal directionality information.

This is closely related to the Reviewer’s public comment #1. The text below is largely repeated for convenience.

We have clarified the result about directional information in the new manuscript.

First, it is not optimized, in the sense that there are other parameters that would give more directional information – we apologize for the lack of clarity. Rather, the parameters observed are such that changing them would either reduce search success or directional information. In the study of multiple optimization, this fact is called “Pareto optimality”.

Second, the prior intuition is that weaker diffusion (straighter airinemes) would provide more directional information. This was indeed our intuition as well, prior to this study. To our surprise, we found that both very weak diffusion and very strong diffusion both give local maxima of directional information. The intuitive explanation is that the searchers are finite-length, and high diffusion leads to a smaller search extent which only reaches the target cell at its very nearest region. We provide this intuitive explanation (which was indeed a surprise to us) in the Results section.

Third, the Reviewer asks about the generality of the result about directional information. This is an excellent question. The comment, and similar comments from other Reviewers, prompted us to perform a parameter exploration study. This is contained in a new Supplemental Figure and new paragraphs in the Results section. See our comments above, repeated here for convenience:

The Reviewer’s question is an excellent question: Is the trade-off between contact and directional information a general property of searchers that obey persistent random walks? To address this question, we now include the analysis previously contained in Figure 5D, but for a full parameter space exploration. This is done in new Figure 5 Supplemental Figure 1. In doing so, we found fascinating behavior that sheds some light on the loop in Figure 5D.

At low d_targ, the trade-off is amplified, and the parametric curve resembles bull's horns with two tips representing the smallest and largest D_theta in our explored range, pointing outward so the shape is concave-up. Intuitively, we understand this as follows: since the target is fairly close (relative to l_max), contact is easy. The only way to get directional specification is by increasing D_theta to be very large, effectively shrinking the search range so it only reaches (with significant probability) the target at the near side (“3-o-clock'' in Figure 5A). At low d_targ, the parametric curve is concave-up, and there is no Pareto optimum.

At high d_targ, the searcher either barely reaches (when D_theta is high), and does so at 3-o-clock, therefore providing high directional information, or D_theta is low, and the searcher fails to reach, and therefore also fails to provide directional information. So, at high d_targ, there is no trade-off.

At intermediate d_targ, the curve transitions from concave-up bull's horn to the no-tradeoff line. To our surprise, it does so by bending forward, forming a loop, and closing the loop as the low-D_theta tip moves towards the origin. At these intermediate d_targ values, the loop offers a concave-down region with a Pareto optimum.

So, to answer the specific question of the Reviewers: No, the Pareto optimum is not a general feature of persistent random walk searchers. It only exists in a particular parameter regime, sandwiched between a regime where there is a strict trade-off with no Pareto optimum and a regime in which there is no trade-off.

All of these results are now discussed in the main text.

(Note that although we do not explicitly explore lmax, since these plots have not been nondimensionalized, the parametric curve for a different lmax can be obtained by rescaling the results).

2) Enhance the discussion regarding the biology of the system. What are you really modelling? The motion of a cellular protrusion whose velocit¥ and persistence is related to its molecular constituent (cytoskeleton) or thge motion of an entire cell (the macrophage quiding the protrusion).

This is closely related to the Reviewer’s public comment #2. The text below is largely repeated for convenience.

These are very good questions. Airinemes have been characterized in a few studies since their discovery in 2015. We are saddened (and excited) to say that: the answers to all of these questions are currently unknown. To paraphrase the Reviewer, the questions are: First, what is the force generation mechanism that leads to airineme extension (additionally, if there are multiple coordinated force generators, e.g., the airineme’s internal cytoskeleton and the macrophage, how are they coordinated)? And second, what are the molecular details of airineme tip contact establishment upon arrival at a target cell?

We present an extended biological background discussion addressing these questions, including what is known and what remains unknown. See response above, repeated here for convenience. We have incorporated a shortened version of this as a new paragraph in the introduction.

Airinemes are produced by xanthophore cells (also called yellow pigment cells) and play a role in the spatial organization of pigment cells that produce the patterns on zebrafish skin. Xanthophores have bleb-like structures at their membrane, and those blebs are the origin of the airineme vesicles at the tip. Those blebs express phosphatidylserine (PtdSer), an evolutionarily conserved ‘eat-me’ signal for macrophages. Macrophages recognize the blebs, ‘nibble,’ and ‘drag’ as they migrate around the tissue and the filaments trailing and extending behind. Airineme lengths have a maximum, regardless of whether they reach their target. If the airineme reaches a target before this length, the airineme tip complex recognizes target cells (melanophores) and the macrophage and airineme tip disconnect.

The airineme tip contains the receptor Δ-C, which activates Notch signaling in the target cell. The mechanism by which a macrophage hands off the airineme tip is still mysterious, due to temporal and spatial resolution limits. It is also known what other signals, if any, are carried by the airineme. If no target cell is found by the maximum length, the macrophage and airineme disconnect, and the airineme the extension switches to retraction. Thus, macrophages do not keep dragging the airineme vesicles until they find the target melanophores. However, how macrophages determine when to engulf the untargeted airineme vesicles is not understood.

In a different context, during wound-healing, macrophages are recruited to the site of injury by detecting chemokines released by damaged cells or other immune cells. However, airineme pulling macrophage behaviors are not triggered by tissue damage or infections in zebrafish skin, and there is no evidence that they respond to any directional cues.

Regarding the Reviewer’s specific question about the implications for the macrophage on how we model contact establishment: This would indeed change the interpretation of the model parameter r_targ. Specifically, contact is declared when the airineme tip arrives at a distance r_targ from the center of the target cell, and this critical distance might be larger than the size of the target cell, since it might include part or all of the macrophage. We have added this to the first part of Results, when the parameter is introduced.

3) Discuss the data in more detail, in particular how well they really agree with the model.

This is closely related to the Reviewer’s public comment #3. The text below is largely repeated for convenience.

The MSD analysis allows us to reject the simple random walk model, and it is consistent but alone is not strongly supportive of the PRW model, especially at high tau of around 15 minutes (long lengths of around 65 microns). As the Reviewer points out, this is due to low numbers of long airinemes.

We agree, and have performed new analysis. The following is repeated here for convenience:

The lack of strong agreement prompted us to investigate the long-length data using multiple analysis approaches. In the new manuscript, new Figure 2B, we took all airinemes whose growth time was greater than 15 min, and plotted their final angle, i.e., the angle between the tangent vector at their point of emergence from the source cell and the tangent vector at their tip. At long times, the PRW model predicts that, for long times >1/D_theta, the angular distribution should become isotropic. In new 2B, we find that the angular distribution is uniform, i.e., isotropic, using a Kolmogorov-Smirnov test (p-value 0.37, N=26).

Since there are relatively few data points, we repeated this analysis under various airineme selection criteria, and in all cases found the final angular distribution to be consistent with uniformity (new Supplemental Data Figure 1). For example, if we set the threshold at 10min, which includes up to N=49 airinemes, the Kolmogorov-Smirnov test against a uniform angular distribution gives a p-value of 0.32.

We here add a few additional notes

– Note that there is significantly less data used in this test than in the MSD analysis or the autocorrelation function maximum likelihood analysis. In order to perform a hypothesis test, we wanted to be sure that the data points are independent, so we take only one from each airineme (unlike MSD and autocorrelation analyses, for which we take every interval of a particular length, whether in the same airineme or not.)

– Finally, although the >10min KS test has more data than the >15min KS test (N=49 compared to N=26), we have chosen to present the >15min KS test in the Main Text. As we mentioned above, the conclusion is unchanged for >10min (see Supporting Data). The reason is that >15min is the first test we ran to check angular distribution against a uniform (-pi,pi) distribution, and we did not want to bias our testing.

Taken together, the data are even more strongly supportive of the PRW model. We are grateful for the Reviewer in encouraging us to further explore the high-time data.

4) Discuss the assumption of the model more precisely, in particular regarding directional information.

This is closely related to the Reviewer’s public comment #4. The text below is largely repeated for convenience.

The sketch in Figure 5 was indeed not clear about generality, so we have edited it so that the angles are no longer perpendicular.

We also now also clarify in the Main Text that, in all simulations (both measuring contact probability and directional sensing), the airineme begins at a specified point in an orientation uniformly random in (-pi,pi). We apologize that this was not clear in the previous sketch.

Regarding hindrance by the source cell: While the tissue surface is crowded, the airineme tips (which are transported by macrophages (Eom et al., 2015)) appear unrestricted in their motion on the 2d surface, passing over or under other cells unimpeded (Eom et al., 2015). We therefore do not consider obstacles in our model. This includes the source cell, i.e., we allow the search process to overlie the source cell. We now state this explicitly in the Main Text.

Regarding two maxima in Figure 5C: We understand it with the following intuitive picture. For low D_theta, i.e., for very straight airinemes, the allowed contact locations are in a narrow range (by analogy, imagine the day-side of the planet Earth, as accessible by straight rays of sunlight), resulting in high directional information. For high D_theta, i.e., for very random airinemes, we initially expected low and decreasing directional information, since there is more randomness. However, these are finite-length searches, and the range of the search process shrinks as D_\theta increases. This results in a situation where the tip barely reaches only the closest point on the target cell, resulting again in high directional information. We have added this intuitive reasoning in the Main Text.

Reviewer #4 (Recommendations for the authors):I have only a few remarks that could be taken into account to improve clarity of the manuscript.In the current version of the paper, one must go to the material method section to understand that there is a maximal length for airinemes. For clarity it should probably be better to mention it in the main text, because it is an important point of the discussion of Section 2.2 and Fig 3A. Indeed it is very well known that a 2D a diffusive walker will always find any target, which makes very surprising the Figure 3A until one understands that there is a maximal length in the model.

The finite length (after which the search process terminates if unsuccessful) is now mentioned and justified in the Introduction and again in the first Results section, referring to supplemental Figure S4. The Reviewer also makes an important statement, that two-dimensional diffusion is recurrent so always successful. We feel this is important to point out. So, in this version, we state this explicitly, and repeat the finite-length property upon first mention of search probability.

- Again for clarity it could be useful to present Fig1B also in semi-log scales since this type of curved lines in log-log scales may simply be exponentials. Identifying an exponential law for the step size distribution would certainly lead to a rejection of Levy type walks.

We now include the same data (the complementary cumulative distribution of the step size) on semi-log axes, as an inset (new figure 2C).

Please also define clearly what is the "best fit exponent" (Section 2.1, first paragraph) : which exponent is it (I guess it is the exponent in the MSD) ? Also what is the step size shown on Fig 2B (is it the distance travelled during a specific time ?)

We have added definitions of both best fit exponent and step size. The Reviewer was correct about both of these.

- To avoid any confusion, it would be useful to draw Fig5A with an airineme that is not perpendicular to the cell surface, so that there is no confusion between the angle that the airineme's tip makes with the cell surface, and the contact angle.

Agreed. Done. In the new schematic, the departure point is also not centered on the source cell, to further reduce confusion.